# Exploring Foveation and Saccade
# for Improved Weakly-Supervised Localization

**Timur Ibrayev**                                            TIBRAYEV@PURDUE.EDU

**Manish Nagaraj**                                          MNAGARA@PURDUE.EDU

**Amitangshu Mukherjee**                               MUKHER44@PURDUE.EDU

**Kaushik Roy**                                              KAUSHIK@PURDUE.EDU

*Purdue University,*
*West Lafayette, IN 47906 USA*

## Abstract

Deep neural networks have become the de facto choice as feature extraction engines, ubiquitously used for computer vision tasks. The current approach is to process every input with uniform resolution in a one-shot manner and make all of the predictions at once. However, human vision is an "active" process that not only actively switches from one focus point to another within the visual field, but also applies spatially varying attention centered at such focus points. To bridge the gap, we propose incorporating the bio-plausible mechanisms of foveation and saccades to build an active object localization framework. While foveation enables it to process different regions of the input with variable degrees of detail, saccades allow it to change the focus point of such foveated regions. Our experiments show that these mechanisms improve the quality of predicted bounding boxes by capturing all the essential object parts while minimizing unnecessary background clutter. Additionally, they enable the resiliency of the method by allowing it to detect multiple objects while being trained only on data containing a single object per image. Finally, we explore the alignment of our method with human perception using the interesting "duck-rabbit" optical illusion. The code is available at: https://github.com/TimurIbrayev/FALcon.

**Keywords:** neuro-inspired algorithms, foveation, saccades, active vision, optical illusions, weakly supervised learning, object localization, object detection, deep learning

## 1. Introduction

Deep Neural Networks (DNNs) have brought a lot of advancements to the field of computer vision (Deng et al., 2009; Krizhevsky et al., 2012). They have become the de facto choice as feature extraction engines, ubiquitously used for downstream tasks of object classification, localization, and detection (Ren et al., 2015; Liu et al., 2016; Redmon et al., 2016; Redmon and Farhadi, 2016; Lin et al., 2017, 2020). However, their nature of processing inputs can be characterized as being "passive", meaning that the entire image is processed in a one-shot manner with uniform resolution (i.e. spatially invariant attention within the visual field), and all of the predictions are made at once. In contrast, empirical observations from neuroscience (Curcio et al., 1990; Eckstein, 2011; Land and Nilsson, 2012) suggest that human vision is an "active" process that not only actively switches from one focus point to another within the visual field, but also applies spatially varying attention centered at such focus points.

To that effect, we advocate incorporating foveation and saccades into machine perception. *Foveation* is the property of the human eye to process different regions of the visual field with variable degrees of detail. *Saccades* are quick eye movements that allow changing

the focus point from one location to another. Their combined implementation can grant computer vision methods the ability to select where to look, ignoring the unnecessary clutter from the input. Such adaptive functionality may not be possible when the entirety of an input is processed with uniform resolution.

Note, foveation and saccades allow different views from within the same sample and hence augments the set of features used for training. Figure 1 illustrates this by comparing the processing (a) without and (b) with the mechanisms of foveation and saccades (collectively denoted as "F&S"). Figure 1(a) shows the standard computer vision pipeline that processes the input with uniform resolution. Training for a downstream task, e.g. the classification of the object or the prediction of its location, depends on the set of features that DNN is able to extract from the set of samples. When the entirety of the inputs is processed with uniform resolution, DNNs have to rely on the views of different samples belonging to the same class (*intra-class inter-sample views*) in order to come up with a consensus on which features describe the object of this class. For example, 1000 different "dog" images allow DNN to learn parts/features of the "dog" class that are necessary to either classify an image as depicting the "dog" class or draw a *complete* bounding box *capturing all of its parts* to predict its precise location.

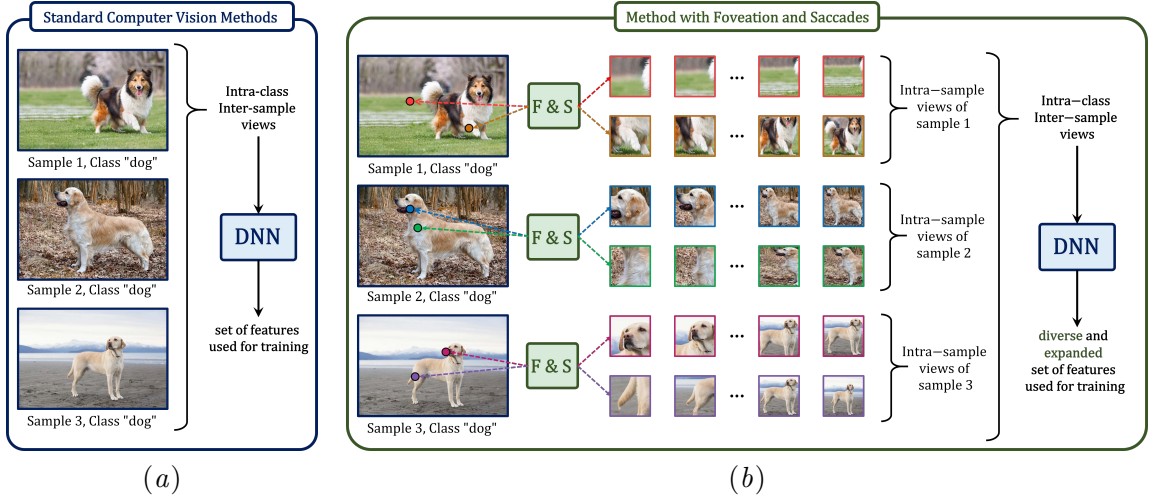

$(a)$ $(b)$

Figure 1: Comparison of the processing pipelines for (a) standard computer vision methods processing inputs with uniform resolution and (b) a method with foveation and saccades. "F&S" denotes the mechanisms of foveation and saccades. They expand the set of training samples by obtaining different views from within each sample at different fixation points (due to saccades) and different degrees of details at these fixation points (due to foveation). Best viewed in color.

On the contrary, Figure 1(b) shows the pipeline incorporating "F&S". With different fixation points (due to saccades) and different degrees of details around these fixation points (due to foveation), "F&S" can result in a large number of views obtained from within a single sample (*intra-sample views*). As a result, DNNs have a larger set of features that it can rely on to come up with the consensus on which features describe the object of this class. Moreover, this expanded set is more diverse as the intra-sample views might contain

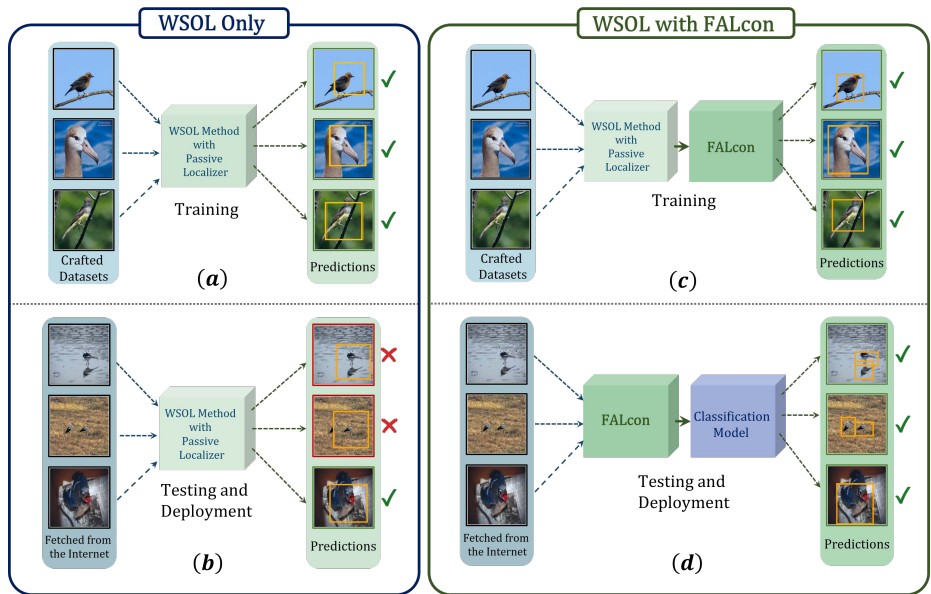

Figure 2: FALcon improves the localization performance and enables the detection of multiple objects while only being trained on the data that contains only a single object. If we assume that the localization is performed in some unconstrained environment, e.g. images are fetched from the internet, FALcon can be thought of as an advanced WSOL method that is more resilient to images with multiple objects during the final deployment.

the entire object, only some parts of the object, or none of the object parts. This might be beneficial for learning fine-grained features in the case of classification tasks and for learning the "complete" object in the case of bounding box prediction tasks. For example, "F&S" might produce 10 views of a single "dog" image, some of which might purposefully drop any of the "dog" class features otherwise observable from the whole image. Then, the DNN can learn that such negative examples are incomplete in comparison with the "dog" shown in the whole image. This forces the DNN not only to learn consistency in intra-class inter-sample features but also in intra-sample features.

Incorporating the mechanisms of foveation and saccades, we propose an (active) object localization framework. They change the task of predicting object locations from being a "passive" process to an "active" process. The core idea is to reformulate the prediction of bounding boxes from a one-shot manner to an iterative process. By emulating *foveation*, our proposed framework observes only the limited (*foveated*) region of the input at each iteration. By emulating *saccades*, the framework is able to predict whether the current foveated region has any relevant object or whether it is better to switch to the next portion of the input. The framework then iteratively predicts the correct sequence of actions to expand the *foveated region* to eventually capture the entirety of the object of interest.

The proposed framework, which we refer to as **FALcon**, was verified within the settings of weakly supervised object localization (WSOL). WSOL is the task that requires prediction of both the object class (*image-level label*) and the object location in the form of a bounding box (*instance-level label*), **while being trained only on the image-level labels**. Since

the degree of supervision is already limited due to the absence of the object locations during training, we show that the addition of foveation and saccades facilitates improved localization. In particular, we combine the proposed FALcon framework with the existing WSOL method (Figure 2) and show that the bio-plausible mechanisms provide the following benefits:

1. We show that replacing standard "passive" localization with the FALcon improves the localization performance on the WSOL task (compare predictions in (a) and (c) of Figure 2). This is achieved due to more complete (i.e. capturing all essential object parts) and/or more tight (i.e. avoiding all unnecessary background clutter) bounding boxes resulting from the proposed utilization of foveation and saccades.

2. We show that FALcon enables a resilient WSOL pipeline. If we assume that the end goal is to have an automated annotation system, where the images might be fetched directly from an unconstrained environment like the internet, it is desirable to have a method that is resilient to images with multiple objects (even if trained only on data containing a single object). Hence, we show that using foveation and saccades it is possible to obtain a model that is capable of detecting multiple objects while being trained on data that contains only a single object (shown in (b) and (d) of Figure 2).

3. Finally, we show that designing neuro-inspired algorithms with mechanisms like foveation and saccades might be a promising research direction. Specifically, we show that, similar to humans, FALcon is capable of providing two different results (based on the starting point of the foveated glimpse) on the well-known "duck-rabbit" optical illusion.

## 2. Background

### 2.1. Foveation and Saccades

The human vision has spatially varying acuity due to a non-uniform topography of photoreceptor cells in the eye's retina layer (Curcio et al., 1990). We broadly refer to such visual processing with variable degrees of detail using the term "foveation". Being in contrast to computer vision that processes inputs with uniform resolution, there has been research interest to explore the benefit of foveation for machine perception. In particular, the works that model and incorporate foveation showed its effectiveness for computational efficiency (Bauer et al., 2023; Jaderberg et al., 2015; Thavamani et al., 2021), image classification (Jaderberg et al., 2015; Pramod et al., 2022; Yang et al., 2022), scene understanding (Wu et al., 2018; Thavamani et al., 2021), object discovery and saliency (Matzen and Snavely, 2015; Yang et al., 2022), image generation (Gregor et al., 2015), and robustness against adversarial inputs (Deza and Konkle, 2020; Gant et al., 2021). Because the extent of the visual field that is processed with high details is limited, the responsibility of guiding it is assigned to "saccades": rapid movements of the eye between multiple fixation points (Eckstein, 2011; Land and Nilsson, 2012). Various works were proposed either to predict eye movements (Jindal and Manduchi, 2023) or to use eye movements (or *gaze* fixations) as the data for object counting (Thompson et al., 2023), maze solving (Li et al., 2023), facial composite generation (Strohm et al., 2023), automated/assisted driving (Nikan and Upadhyay, 2023), and

robot guidance (Yifan et al., 2023). Finally, the combination of both foveation and saccades has been shown useful for the applications of visual search and guidance. Cheung et al. (2017) demonstrate the importance of retinal sampling lattice for visual search task of an object in a cluttered scene using a neuron attention model. Akbas and Eckstein (2017) aggregates observations across multiple fixation points via a Foveated Object Detector as an alternative to the sliding window approach of passive detectors. Zhang et al. (2018a) searches for a target in a cluttered scene using biologically inspired computation model with zero-shot training. The works of Elsayed et al. (2019) and Huang et al. (2022) propose different combinations of hard attention and reinforcement learning to efficiently improve the classification performance. We propose a complementary approach of utilizing foveation with extreme cutoff as the method of hard attention and saccades as the method of estimating the relevance of foveated observations for the purpose of advancing the weakly supervised object localization methods.

### 2.2. WSOL: Weakly Supervised Object Localization

In this work, we advocate the efficacy of the novel application of foveation and saccades in the task of WSOL. Here we briefly describe some selected works with which we compare in the Results section and later provide an extensive study of previous WSOL methods in the Appendix. Zhou et al. (2016) re-purpose the global average pooling layers of convolutional neural networks to generate class activation maps (CAM) to obtain localization maps. However, CAM highlights only the most discriminative parts of an image leading to the part dominance issue which hinders localization estimates of DNNs. SPG (Zhang et al., 2018c) separates the foreground from the background by learning spatial correlation information among pixels. PSOL (Zhang et al., 2020a) encourages that weakly supervised object localization should be divided into two parts: a localization network performing class-agnostic object localization and a separate classification network performing class-specific object classification. SLT-Net (Guo et al., 2021) employs a separate localizer to learn the localization function by matching the class activation map of the input image and the inverse activation map of a transformation of the same image. $C^2AM$ (Xie et al., 2022) learns a class-agnostic activation map via a novel cross-image foreground-background contrastive loss to disentangle foreground from background on the assumptions that similar foreground objects have similar semantic feature representations which differs from that of the backgrounds. Existing methods provide object locations by directly predicting bounding box dimensions from the single uniform processing of the input. Our work advocates that transforming such "passive" processing into "active" iterative processing further boosts their localization performance and enables their resiliency as an end system.

## 3. Methodology

### 3.1. Active Localization with Foveation and Saccades

The idea behind FALcon is to train a DNN model to predict the location of an object by iteratively observing different parts of the input image and regulating the size of these observations. To be precise, FALcon predicts the location of the object by: (a) focusing on different locations on the image (which we refer to as *initial fixation points*) using

saccades, and (b) iteratively observing only parts of the input image (which we refer to as **foveated/foveation regions**), which are initiated at the fixation points but are expanded as necessary using foveation.

Figure 3(c) illustrates the processing of a single image using different fixation points and different sequences of foveated regions originating from the corresponding fixation points. The top row shows two different initial fixation points indicated by the small circles placed on the full-scale inputs. The two columns of smaller patches under the first row show the two different sequences of foveated regions, which originated from the corresponding initial fixation points. Specifically, the location of an initial fixation point serves as the center for an initial foveated region. Then, the initial foveated region is obtained by cropping out a portion of the input image of a pre-determined size, the value of which is chosen as one of the hyperparameters. For example, the initial foveated region resulting from the first initial fixation point is shown as the first crop under the first image in the top row. FALcon operates by processing *only such foveated regions*, i.e. only the limited regions cropped out from within the input, but not the full-scale input.

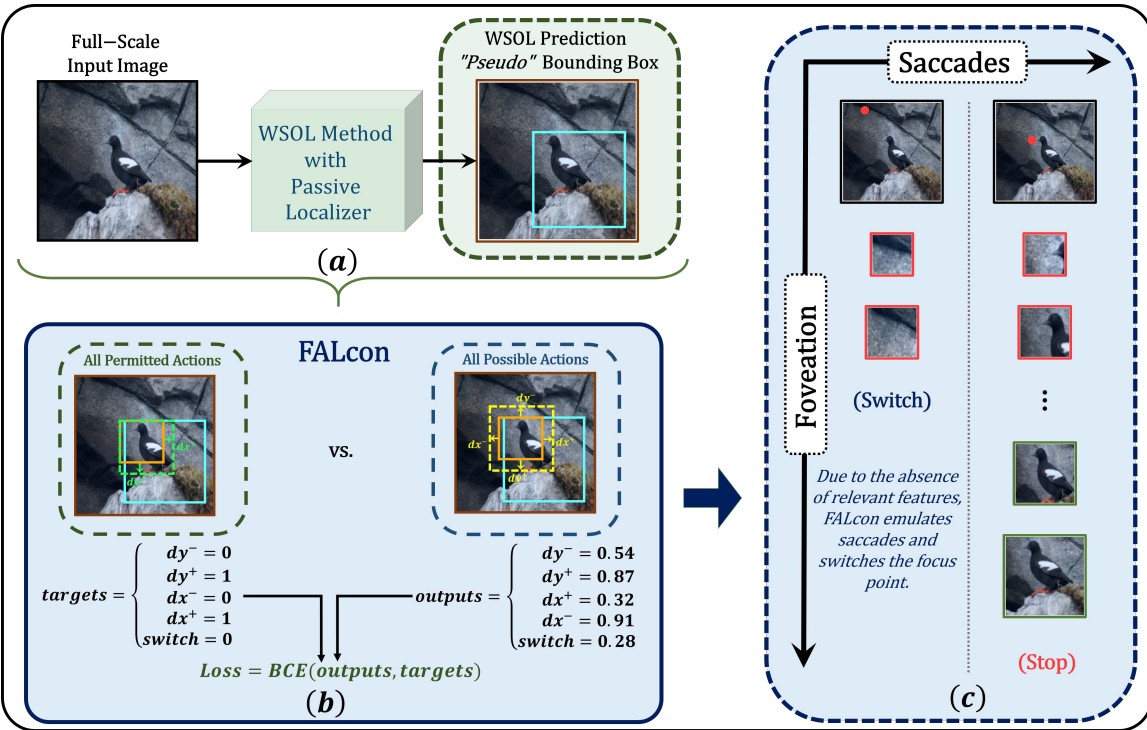

Figure 3: High-level overview of FALcon for weakly supervised object localization (WSOL) task. (a) In the context of WSOL, during training, we first extract predicted bounding boxes using the underlying WSOL method, which are referred to as "pseudo" bounding boxes. (b) FALcon actions are trained based on the current foveation region (orange box) and "pseudo" bounding boxes (cyan box). All possible actions (yellow dashed box) are trained on the set of permitted actions (green dashed box) using binary cross entropy loss. (c) Example illustrating the working principle of active localization based on foveation and saccades mechanisms.

The process of expanding the foveated regions and changing the initial fixation points is learnt by FALcon using a single DNN model with 5 output nodes. By observing the current foveated region, FALcon produces 5 outputs, each corresponding to one of the five allowed actions for the localizer to choose from. Four actions emulate the foveation mechanism that controls the size of the foveation region. In particular, the four actions are: (1) expand in $(dy^-)$ direction, (2) expand in $(dy^+)$ direction, (3) expand in $(dx^-)$ direction, (4) expand in $(dx^+)$ direction, with $(0,0)$ coordinates being top left corner of the full-scale input image. The last action ((5) switch) emulates the mechanism of saccades that controls whether the current foveation region contains anything of interest or the region is irrelevant and the fixation point needs to be switched. The predicted actions at each iteration are then applied in an iterative manner to either change (expand) the current foveated region or to completely switch the fixation point. As a result, a successful localization is achieved when FALcon decides neither to expand the foveation region nor to switch the fixation point to investigate any other portion of the input.

Figure 3(c) illustrates one possible course of actions that FALcon can take to localize the bird in the image. From the first initial fixation point, FALcon expands the initial foveated region to observe a larger portion of the input (as shown by the left column of Figure 3(c)). However, due to the absence of any relevant features in the initial and the second foveated regions, FALcon switches to the next fixation point. From the second initial fixation point (shown in the right column), FALcon produces and observes a series of expanding foveation regions. Finally, the bird in the input image is successfully localized by FALcon capturing it within the last foveated region and predicting no further action.

### 3.2. Training Approach

The proposed training approach simultaneously uses different views generated by foveation and saccades and trains FALcon to improve them for the purpose of object localization. This is achieved by allowing FALcon to generate a sequence of foveated regions and training it on each of the foveated regions with the goal of capturing the object of interest. For each image, the sequence of observing and expanding foveated regions lasts for a pre-determined number of iterations (a hyperparameter referred to as the *foveation iterations*). As FALcon becomes better at capturing relevant object parts, the relevance of the foveated regions to the task improves. Due to different initial fixation points, FALcon has to learn to capture objects using various trajectories of foveated regions, which ensures the diversity of foveation iterations. For example, compare the foveation iterations to localize a "dog" in sample 3 from its "head" versus from its "tail" shown in Figure 1(b). As a result, the training progression itself increases the number of diverse and task-relevant training samples.

Figure 3(b) shows the training approach of the FALcon model at one of the foveation iterations. The foveated region at the current iteration is shown as an orange bounding box. On the right part of Figure 3(b), we can observe FALcon expand the current foveated region (orange bounding box) into all four directions (dashed yellow bounding box). FALcon is designed to take each expansion with a pre-determined fixed step size, which is one of the hyperparameters. In other words, if one of the edges is (predicted) to be expanded, it is changed by the fixed number of pixels. Otherwise, the foveated region dimensions will remain unchanged. We represent the expansion of the right, the left, the bottom, and the

top edges of the foveated region as $(dx^+)$, $(dx^-)$, $(dy^+)$, and $(dy^-)$, respectively. The model predicts the output values for each expansion direction, each representing the confidence of the model (in the range $[0, 1]$) to expand the corresponding edge of the foveated region. The predictions for each edge are independent, meaning that at each iteration, the model can decide to expand in all four directions or keep the entire region unchanged.

Training of FALcon requires a reference location of the object of interest, shown as the solid cyan bounding box in Figure 3(b). The targets to train the FALcon actions are generated based on the current foveated region and the reference box. Specifically, the target value for each edge is true, if the edge of the new (expanded) foveated region does not exceed beyond the dimensions of the reference box. Otherwise, the target is assigned a false value which directs FALcon not to expand in that direction. For the example shown in Figure 3(b), we can see that out of the 4 possible expansion actions only two are allowed, which will make the foveated region at the next iteration (dashed green bounding box) remain within the boundaries of the reference bounding box (shown in solid cyan).

Along with the foveation actions, FALcon is also trained to learn the saccades mechanism by estimating the relevance of the current region and determining if a switch of the fixation point is necessary. During training, this is achieved by purposefully initiating fixation points outside the reference bounding box for half of the images and within the reference bounding box for the other half. For the first half of the samples, the switch target is true, until the model correctly predicts the switch action. If the model predicts the switch action correctly, it is provided with a new initial fixation point that is within the reference bounding box. For the samples that have the fixation point within the reference bounding box (i.e. for the second half of the samples as well as the ones that successfully predicted the required switch), the switch target is changed to false.

Note that the behavior of switch predictions by saccades is different from the prediction of "objectness" scores in the standard computer vision detectors. Standard detectors assign the objectness score based on the final overlap of the predicted bounding boxes with the ground truth bounding box. As a result, while standard detectors need to learn to differentiate foreground and background implicitly through multiple predicted boxes and their assigned objectness scores (similarly to multiple instance learning), our method trains the difference explicitly by directly sampling portions of the image pertaining to the background.

### 3.3. FALcon Deployment

From the application perspective, we consider FALcon as a replacement for the standard "passive" localization (layers or an entire model) that produces the final coordinates of the predicted bounding box after a single observation with uniform resolution. As explained in the previous subsection, FALcon is a supervised technique and requires at least a rough estimate of the object location. In the context of weakly supervised object localization, FALcon hence builds upon an existing WSOL method. Specifically, in order to train FALcon, a pre-trained WSOL method is used with its standard localization model to estimate the object location for the given input training image (Figure 3(a)). We refer to these bounding boxes predicted by the underlying WSOL method as "pseudo" bounding boxes in order to highlight the difference between ground truth bounding boxes used for final evaluation. After training is complete, FALcon works on its own without a need for the

localization part of the underlying WSOL method (as shown in Figure 2(d)). For a fair comparison, however, we still take the same classification model of the underlying WSOL method. Although we did not consider FALcon in the context of fully supervised object localization and detection tasks, the proposed method should still be applicable when the ground truth bounding boxes are available for training.

In order to capture all the possible objects, during inference FALcon explicitly considers multiple initial fixation points equally distributed over the input image. This is essentially similar to dividing the entire image into grid cells and considering each of the cells as a starting foveation region. However, all of the grid cells that were predicted to switch are dropped from consideration. All of the remaining final foveated regions (which essentially represent predicted bounding boxes) are then passed through a non-maximum suppression technique, similar to most object detectors. This approach allows the proposed framework to manifest the resiliency that it learns during training. In particular, the learned saccadic switch behavior allows FALcon to consider and explore any place in the input space that potentially has relevance to any of the objects of interest. Additionally, the learned foveation behavior allows FALcon to expand and stop foveated regions to capture such objects. Such ability is neglected in object localization methods that process inputs with a uniform resolution, due to the underlying assumptions that only a single object is present and the resiliency is not necessary.

## 4. Results

### 4.1. Experimental Settings

**Implementation** The proposed method was implemented using the PyTorch deep learning framework. The training and evaluation experiments were performed on a single Nvidia A40 80GB GPU card. In the context of the WSOL task, we used the PSOL work (Zhang et al., 2020a) as the underlying WSOL method to provide "pseudo" bounding boxes for training images. Then, the proposed framework is trained on "pseudo" bounding boxes extracted using the PSOL method on all training images. FALcon relies on the number of hyperparameters. A detailed description of hyperparameters is provided in the Appendix.

**Datasets** We trained the proposed method on two datasets in WSOL settings: Caltech-UCSD Birds-200-2011 (Wah et al., 2011) (also known as CUB-200-2011 or CUB) and ILSVRC 2012 (Russakovsky et al., 2015) (also known as ImageNet-1k). CUB-200-2011 is the dataset of 5,994 training and 5,794 testing images belonging to 200 different bird species, which are treated as image-level object classes. ImageNet-1k is a large dataset of 1,281,167 training and 50k validation images, with each assigned one image-level object class label. Note that ImageNet-1k images might contain multiple objects, and its annotations provide multiple bounding boxes. However, during training, it is always assumed that the images contain only a single object described by the provided image-level label. Correspondingly, FALcon is only trained on a single "pseudo" bounding box per image predicted by the PSOL baseline for every image in the training set. To verify the effect of active localization using foveation and saccades, trained FALcon models are evaluated on test and validation images of CUB and ImageNet-1k datasets, respectively. To verify the resiliency of the pipeline with foveation and saccades, we also test the capabili-

ties of the baseline models and FALcon models on multi-object datasets. Specifically, the FALcon model that was trained on the CUB dataset was verified on Pascal VOC07 and Pascal VOC12 datasets (Everingham et al., 2010), and the FALcon model trained on the ImageNet-1k dataset was verified on the detection set of ILSVRC2013 (Russakovsky et al., 2015) (which we refer to as ImageNet13-Det).

**Evaluation**   To show the effect of foveation and saccades both on the localization performance as well as the resiliency, FALcon is evaluated on images containing only a single object (i.e. CUB and ImageNet-1k datasets) and on images containing multiple objects (i.e. Pascal VOC07, Pascal VOC12, and ImageNet13-Det). (*Note: for both cases, FALcon is trained only on images containing a single object.*) Hence, the following evaluation metrics were used. In the context of the localization of a single object, our method is evaluated on "GT-Known Loc" and "Top-1 Loc" metrics. "GT-Known Loc" or "GT Loc" is the percentage of samples for which the intersection over union (IoU) between the ground truth bounding box and the predicted bounding box is greater than or equal to 0.5 (i.e. localization only). "Top-1 Loc" is the percentage of samples for which both the IoU is greater than or equal to 0.5 **and** the label of the object within the bounding box is correctly predicted (i.e. localization and classification). In the context of the detection of multiple objects, our method is evaluated on "Average Precision (AP)" and "mean Average Precision (mAP)". Both of the metrics measure the area under the curve in the precision versus recall curve, but with "AP" measuring it for a specific class and "mAP" measuring the mean of the AP values for different classes.

### 4.2. FALcon Improving Localization

Table 1 shows the performance of FALcon on WSOL benchmarks in comparison with PSOL (Zhang et al., 2020a) as the baseline and other WSOL methods. In contrast to the standard WSOL methods, FALcon is designed to automatically consider the possibility of multiple objects being present in the image. It is expected that the method will make a single prediction per image whenever there is a single object. Consequently, without explicit supervision or forced restrictions, for the CUB and ImageNet-1k datasets, FALcon produced only a single prediction per image in 94.8% and 71.0% of test samples, respectively. However, there are some test samples for both datasets in which FALcon chooses a safer approach of making more than 1 prediction. For example, for CUB and ImageNet-1k, FALcon made 2 predictions per image in 4.8% and 19.5% of test samples, respectively. More detailed statistics are provided in the Appendix.

Hence, we report GT Loc and Top-1 Loc values for three different scenarios in Table 1: when we consider up to 1, up to 3, or up to 5 predictions per image. For each scenario, we evaluate up to a specified number of predictions per image, ranked based on the confidence of the classification prediction. Then, the object is considered to be correctly localized if any of the predictions correctly predicts the location and the object label.

On the CUB dataset, FALcon achieves significant localization results (based on GT Loc), even when it is limited to making a single prediction per image (max 1). The accuracy of bounding boxes improves from 77.41% of the baseline (PSOL) method to 88.30% with the addition of foveation and saccades. This improvement is explained by qualitative results shown in Figure 4(a). Comparing the bounding boxes produced by PSOL alone and

Table 1: Weakly supervised object localization (WSOL) results

| Method | # of predictions per image | CUB | | ImageNet | |
|---|---|---|---|---|---|
| | | GT Loc | Top-1 Loc | GT Loc | Top-1 Loc |
| VGG16 CAM (Zhou et al., 2016) | 1 | 57.96 | 36.13 | 59.00 | 42.80 |
| InceptionV3 SPG (Zhang et al., 2018c) | 1 | 60.50 | 46.64 | 64.49 | 48.60 |
| VGG16 SLT-Net (Guo et al., 2021) | 1 | 87.60 | 67.80 | 67.20 | 51.20 |
| DenseNet161 $C^2$AM (Xie et al., 2022) | 1 | 94.46 | 83.28 | 68.20 | 59.28 |
| **PSOL (baseline) (Zhang et al., 2020a)** | **1** | **77.41** | **63.56** | **66.28** | **55.31** |
| **FALcon + PSOL (Ours)** | max 1 | 88.30 | 62.82 | 62.45 | 49.39 |
| | max 3 | 89.35 | 63.50 | 67.38 | 53.31 |
| | **max 5** | **89.35** | **63.50** | **67.51** | **53.50** |

\* *Note: official PSOL repository does not contain pre-trained weights on CUB dataset. We report results achieved by training the models with hyper-parameters and methods described in the original work.*

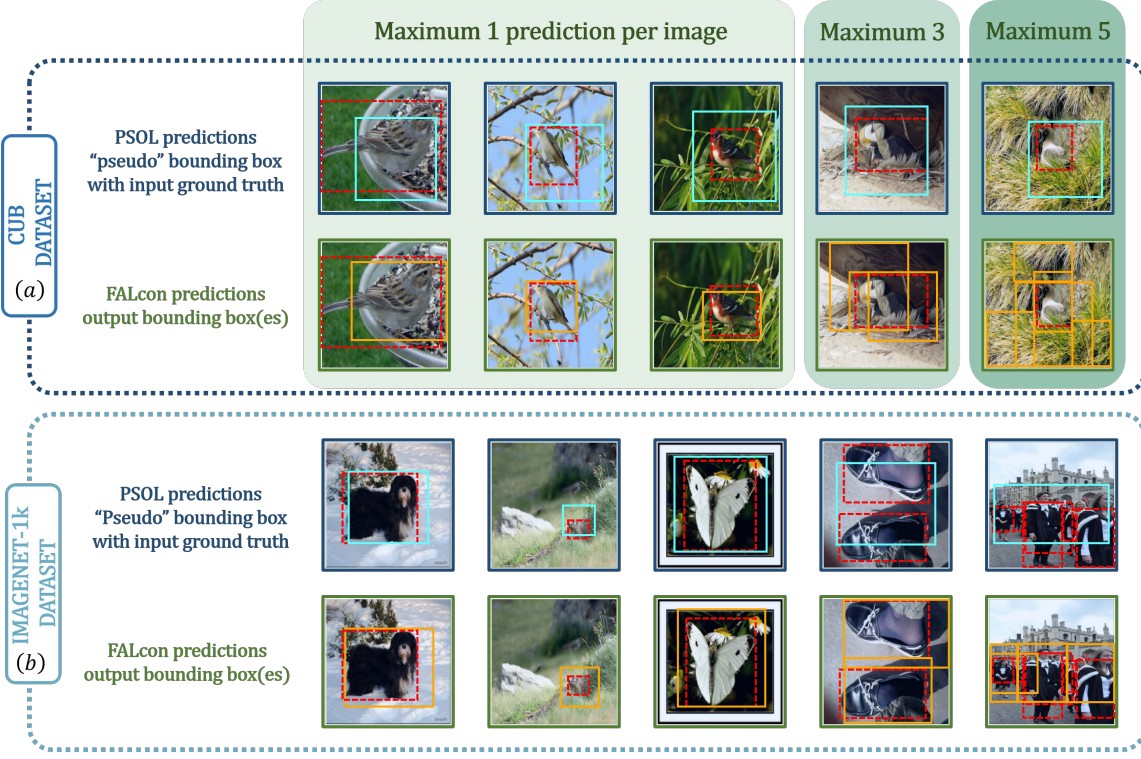

Figure 4: Qualitative results obtained on the (a) CUB and (b) ImageNet-1k datasets.

FALcon trained with PSOL, we can observe that our technique improves ***completeness*** (by capturing more of essential object parts) and/or ***tightness*** (by minimizing the amount of background clutter) of predicted bounding boxes. The capability of FALcon to predict

more than one object per image is not essential for the CUB dataset. Indeed, allowing more predictions results in a small improvement of localization accuracy at the cost of having false positive bounding boxes.

Contrary, on the ImageNet-1k dataset, the capability of FALcon to preemptively make multiple predictions adds a significant benefit. Even when the image-level labels describe a single object class per image, ImageNet-1k is a large and diverse dataset that can contain multiple objects in the image. As a result, the baseline PSOL method fails when there are multiple objects by attempting to find the optimal location that will capture all of them, as indicated by qualitative results shown in Figure 4(b). However, FALcon is not only able to focus on one of the individual ground truth bounding boxes but is also able to make multiple predictions for images containing multiple objects. While this allows the improvement over the baseline method, making the combined approach competitive with other methods, it also motivates an important concern of resiliency to the possibility of multi-object images. When compared with $C^2AM$ method, we believe that their high quantitative performance is attributed to positive and negative pairs used in contrastive learning, which can also be thought of as train-time intra-sample sampling. However, unlike foveation and saccades, which are part of the entire inference pipeline in FALcon, their technique is only used during training, still leaving the final deployed model vulnerable to the presence of multiple objects.

Note that, in this work, we focused on the localization model and its training on the different views generated by foveation and saccades. We did not consider fine-tuning the classification model on the resulting foveated regions. Instead, we applied the same classification model used by the baseline PSOL method with the aim of consistency. As a result, there is a slight degradation in Top-1 Loc accuracy compared to the baseline PSOL method. The reason is that, in contrast to all WSOL methods which process an entire image to predict the label, FALcon makes a classification prediction by only processing the portion of the input within the last foveated region. We believe this is tied to one of the fundamental debates about DNNs (Xiao et al., 2021), whether the background affects a DNN's classification capabilities. In the future work, this can possibly be mitigated (and potentially even improved) by fine-tuning the classification model on the sequence of foveated regions.

### 4.3. FALcon Enabling Resiliency

In addition to qualitative results on ImageNet-1k, we also explicitly verified if foveation and saccades offer resiliency. Specifically, the localization models trained on the CUB and ImageNet-1k datasets by the baseline PSOL method and FALcon approach were applied to multi-object datasets. Models trained on "birds" of the CUB dataset were applied to Pascal VOC07 and Pascal VOC12 datasets and models trained on the ImageNet-1k dataset were applied to the ImageNet13-Det dataset. Table 2 illustrates the performance of detecting multiple objects while being trained only on the data containing a single object per image. While only being trained on a single bounding box per image, FALcon offers higher performance in detecting multiple objects than the baseline PSOL method. Qualitative results shown in Figure 5 demonstrate that foveation and saccades enable the capability to correctly make more than one prediction per image. This is especially noticeable in the case of detecting "birds" in Pascal VOC datasets. The less significant quantitative improvement in detecting ImageNet13-Det objects can be explained by the vast diversity of classes.

Table 2: Results of applying localization models trained on images containing a single object to datasets containing multiple objects per image

| Method | Birds | | | All ImageNet Classes | |
|---|---|---|---|---|---|
| | Training Dataset | Testset (VOC07) $AP_{@0.5}$ | Testset (VOC12) $AP_{@0.5}$ | Training Dataset | Testset (ImageNet13-Det) $mAP_{@0.5}$ |
| PSOL (baseline) (Zhang et al., 2020a) | CUB | 0.32 | 0.42 | ImageNet-1k | 9.89 |
| **FALcon + PSOL (Ours)** | **CUB** | **12.56** | **7.01** | **ImageNet-1k** | **10.32** |

Figure 5: Qualitative results obtained on the (a) VOC07 and (b) ImageNet13-Det datasets.

## 4.4. Towards Neuro-inspired Algorithms

The addition of bio-plausible mechanisms of foveation and saccades was motivated by the idea of bridging the gap between computer vision and human perception. Hence, in order to demonstrate the neuro-inspiration, we applied FALcon model trained on ImageNet-1k to one variation of the famous "duck-rabbit" optical illusion. The interesting aspect of that optical illusion is that human vision allows us to only see one object: either the duck or the rabbit. Due to the nature of our vision, not only are we unable to see both objects at the same time, but we also ignore the features of the other object when focusing on the features of the first. Figure 6 illustrates the result of applying our neuro-inspired approach to this optical illusion. In agreement with human perception, FALcon observes **and** focuses on objects of different classes based on the different fixation points. This exciting finding

motivates the exploration of neuro-inspired algorithms as a plausible research direction for the next generation of machine perception.

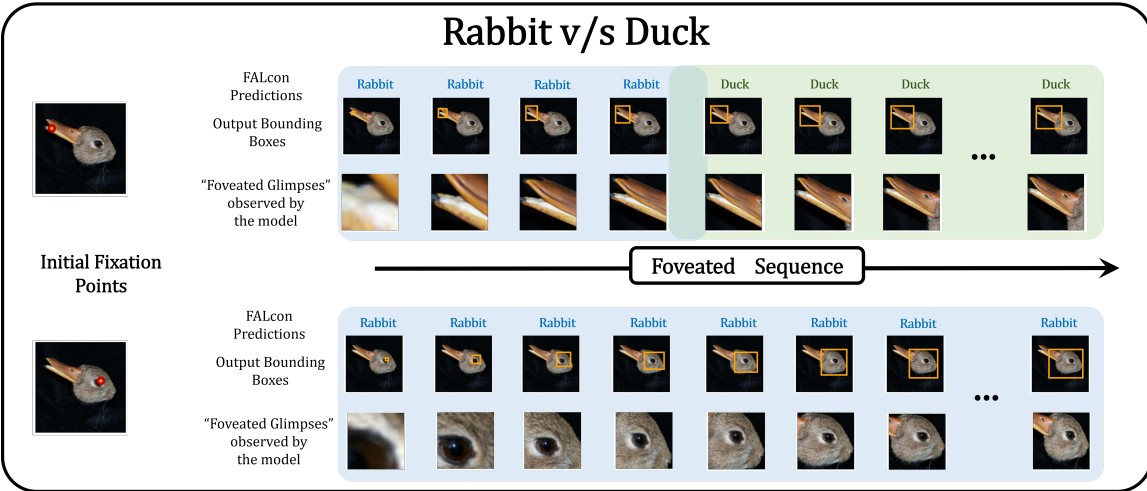

Figure 6: Results of applying our neuro-inspired approach on one variation of the famous Duck and Rabbit optical illusion. Similar to human vision, FALcon observes **and** focuses on objects of different classes based on different fixation points.

## 5. Discussions

**Why F&S improves "pseudo" bounding boxes?** Figure 7 shows examples of foveated regions that can be observed by FALcon during training. Figure 7(a)-(c) are foveated regions generated on the sample with the accurate "pseudo" bounding box, meaning that the underlying WSOL method provides the bounding box that matches the foveated region shown in Figure 7(b). Based on such "pseudo" bounding box, FALcon weights are reinforced for the correct expansion from foveated region shown in (a) to foveated region shown in (b), but punished for the wrong excessive expansion from (b) to (c). As a result, this knowledge is then transferred to improve samples with "pseudo" bounding boxes matching foveated regions shown in Figure 7(d) and Figure 7(e), with FALcon completing foveated region shown in (d) and tightening foveated region shown in (e).

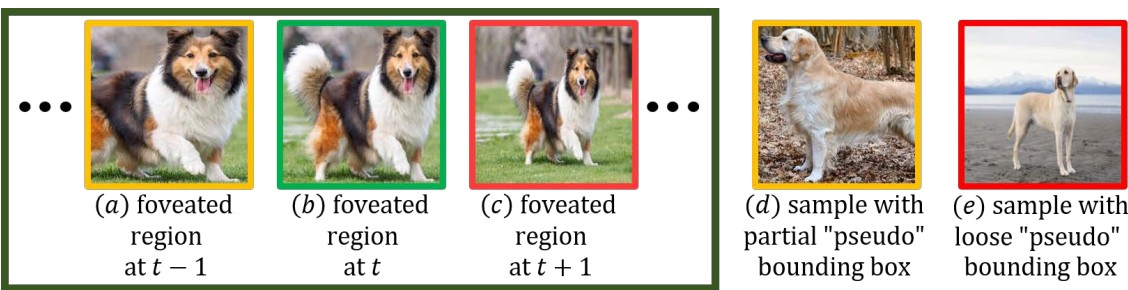

Figure 7: Examples of foveated regions for samples with (a)-(c) accurate, (d) partial/incomplete, and (e) loose/overextended "pseudo" bounding boxes.

**Why consider resiliency and how it is achieved?** The major motivation for WSOL works is to facilitate automated annotation system. The end goal of such systems would be to process images fetched directly from an unconstrained environment, like the internet, without extra human supervision. As such, it is important to consider the resiliency of such systems, i.e. their capability to either flag unexpected input or to detect multiple objects (even if trained only on data containing a single object).

The processing of the input with uniform resolution extracts all the features from the image at once, making it up to DNN to differentiate multiple objects, including those belonging to the same class. In contrast, the resiliency of FALcon is based on the inference of individual foveated regions originating from different fixation points within the input image. Consequently, FALcon extracts explicitly disentangled features, which allows to account for multiple objects.

**Why foveation with extreme cutoff?** In the context of neuroscience and human vision, foveation is a spatially varying attention within the visual field (Larson and Loschky, 2009; Loschky et al., 2019; Rosenholtz, 2016). As a result, previous works on foveation generally implement it using spatially varying operators, similarly to adaptive blur (Larson and Loschky, 2009; Wang and Cottrell, 2017; Pramod et al., 2022) and texture-based modifications (Rosenholtz, 2016; Deza et al., 2018). The application of such operators transforms the importance of different input bits, but still leaves the information available when DNNs process the entire input. Contrary, we wanted foveation to function as explicit hard attention within the input. This allows the processing of individual input regions, while enabling the model to ignore the rest of the image. Hence, the design choice was to implement foveation as extreme cutoff using cropping of the input regions and resizing to a fixed size.

## 6. Conclusion

Motivated by the idea to bridge the gap between human perception and computer vision, we explored the potential of the mechanisms of foveation and saccades to the task of object localization. The resulting active object localization framework, referred to as FALcon, models foveation as the method of observing isolated image regions and saccades as the method of switching fixation points. By allowing these mechanisms to explicitly sample diverse views from within each input, the proposed framework enriches the set of features otherwise limited only to the features obtained from the uniform processing of training images. The capabilities of FALcon were verified within the settings of weakly supervised object localization, where foveation and saccades balance out the weak supervision provided only in the form of a class label for each training image. Our experiments show that FALcon improves upon the localization of the baseline by completing and tightening the bounding box predictions and enables the localization to become resilient to multi object images. Finally, with foveation and saccades, FALcon was able to detect different possibilities stemming from a single duck-rabbit optical illusion example. Similar to humans, based on the initial fixation points, FALcon detects either the duck or the rabbit, but not both at the same time.

## Acknowledgments

The research was funded in part by Center for the Co-Design of Cognitive Systems (Co-CoSys), one of seven centers in JUMP 2.0, a Semiconductor Research Corporation (SRC) program co-sponsored by DARPA, and in part by the National Science Foundation.

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

## Appendix A. FALcon Hyperparameters

In addition to standard hyperparameters determining the neural network architecture and the training process, the proposed method relies on the number of hyperparameters that are inherent to the realization of FALcon with the mechanisms of foveation and saccades. This section introduces each of them and describes their roles. Their exact values are then presented in the next sections along with the other standard hyperparameters specifically for each of two training datasets and three evaluation-only datasets.

*num_glimpses* defines the number of foveation iterations that FALcon takes at each initial fixation point. After an initial fixation point is predicted or the fixation point was switched, the method is forced to predict the correct sequence of actions for this specified number of iterations. This should include both (a) expanding the foveated region until the object of interest is captured and (b) keeping the foveated region unchanged after it was captured.

*glimpse_size_init* serves two purposes. First, it defines the size of the initial foveated region centered at an initial fixation point after the point is initialized or changed. Second, this parameter defines the size of grid cells which divide the entire image. For example, if the image size is $200 \times 200$ and *glimpse_size_init* $= 20 \times 20$, then we would have in total 100 grid cells. Then, the centers of these grid cells are considered as potential initial fixation points.

During training, these fixation points are used to train switching behavior of saccades mechanism. Specifically, all resulting 100 fixation points are sorted into (a) the set of fixation points within the reference bounding box and (b) the set of fixation points outside the reference bounding box. This is performed for every image in the training batch, since different images will have their corresponding reference bounding box. Then, the batch of training images is randomly divided into two categories. For the first category of images in the batch we intentionally give a random fixation point outside the reference bounding box as the initial fixation point. For the second category of images in the batch a random fixation point within the reference bounding box is provided as the initial fixation point. As a result, FALcon learns to predict the saccadic switching behavior by learning to differentiate between the images in the first category requiring the switch and the images in the second category not requiring the switch. After FALcon correctly predicts to switch fixation point for any image in the first category, it is provided with a random fixation point within the reference bounding box. Hyperparameter *ratio_wrong_init_glimpses* determines the ratio of images for which the fixation point is chosen to be outside the reference bounding box. It is set to 0.5 for training on all datasets, splitting the batch during training into half.

During evaluation, FALcon is applied at all 100 fixation points, meaning that each *glimpse_size_init* $= 20 \times 20$ grid cell is considered as the initial foveated region. If at any of *num_glimpses* iterations FALcon predicts the switch action, we discard the fixation point and any resulting sequence of foveated region regardless of how much it expanded. As a result, only the final foveated regions that were not discarded by the switch action are considered as the potential object locations in the image. Note: this can be implemented as either the parallel processing or the sequential processing. The parallel processing would be less accurate in terms of bio-plausibility, but allow for faster computation. The sequen-

tial processing would be more bio-plausible, similar to the human eyes scanning different portions of the input in search of relevant parts.

*glimpse_size_step* defines the step size by which the foveated region is expanded, if FALcon predicts to expand any of its four sides. Since different images in the batch will have foveated regions of different dimensions after the first iteration, in order to enable batch training we reshape all of the extracted foveated regions for all of the images to a fixed predetermined size. Specifically, after extracting the portions of images within their corresponding foveated regions, we reshape all of them to the fixed size defined by *glimpse_size_fixed* and then fetch the batch of reshaped foveated regions to the FALcon DNN model.

We train each of the five actions using binary cross entropy with sigmoid function applied to DNN outputs (see Figure 3(b)). During evaluation, to determine the final predicted actions (i.e. whether to expand any of the foveated region borders or to switch the fixation point), we apply sigmoid function and then threshold the outputs of the trained model. *glimpse_change_th* and *switch_loc_th* define the thresholds for the expansion actions in four directions and the switch action, respectively.

The values of certain hyperparameters were chosen at random based on the general expectation of their behavior. For example, *num_glimpses* was randomly chosen to be 16 for all datasets. This way the model has enough iterations to capture the object and excess iterations to learn to keep the foveated region unchanged after it is captured. The selection of other hyperparameters was performed solely based on the qualitative examination of (the sequence of) foveated regions produced for small number of samples from the training set (at most 30 samples for each training dataset).

## Appendix B. WSOL Experiments

### B.1. CUB dataset

One of the datasets that FALcon is trained on is CUB-200-2011 dataset. First, we obtained reference "pseudo" bounding boxes on the training set using PSOL (Zhang et al., 2020a) as our underlying WSOL method. Despite the original PSOL paper presented results on CUB dataset, the models had to be trained from scratch as the official GitHub repository did not have pre-trained models for this dataset. We trained PSOL method with VGG16 (Simonyan and Zisserman, 2015) architecture for the localization model and ResNet50 (He et al., 2016) architecture for the classification model for CUB dataset. The choice of VGG was motivated by our choice to build FALcon using VGG-type architectures. We attempted multiple training iterations with the training parameters described in the original work and their slight variations. Despite our best attempts to exactly match the results presented in the original work, we settled on the best performing model trained with the exact hyperparameters described in the original work. The results of applying these PSOL trained models to the test set of CUB dataset are presented in Table 1.

Second, we trained our FALcon localization model on bounding boxes predicted by the PSOL method for training set images. For the CUB dataset, FALcon was implemented using VGG11 (Simonyan and Zisserman, 2015) DNN model initialized from ImageNet pretrained weights. The model was trained using SGD optimizer with batch size of 32 samples, 0.9 momentum, 0.0001 weight decay for 100 epochs. Learning rate started from 0.01 and

divided by 10 every 30 epochs. During training, FALcon hyperparameters were set to be: $num\_glimpses = 16$, $ratio\_wrong\_init\_glimpses = 0.5$, $glimpse\_size\_init = (20, 20)$, $glimpse\_size\_fixed = (96, 96)$, $glimpse\_size\_step = (20, 20)$, $glimpse\_change\_th = 0.5$, and $switch\_loc\_th = 0.5$. Here, hyperparameters with two-dimensional values represent the value in $(x, y)$ coordinates. For example, $glimpse\_size\_step = (20, 20)$ means that the foveation region is expanded by 20 pixels in each of $x$ directions (i.e. if $(dx^+) = 1$ or $(dx^-) = 1$) and in each of $y$ directions (i.e. if $(dy^+) = 1$ or $(dy^-) = 1$). For the images resized to $256 \times 256$, these hyperparameter values allowed FALcon to capture the entire image, in case the object of interest fully occupies the image.

During evaluation, the majority of hyperparameters remained the same. However, we had to adjust $glimpse\_size\_init = (40, 40)$ and $switch\_loc\_th = 0.2$. The change to $glimpse\_size\_init$ was motivated by the fact that CUB dataset had very uniform background: most of the images contain either purely blue background representing water or sky, or uniform green background representing trees or forest. It does not affect the quality of predictions, but rather allows the network to make faster decision whether current fixation point should be investigated further. The change to $switch\_loc\_th$ was made based on the qualitative observation of how many samples remain for the final consideration. Recall that we only consider final foveated regions which were not discarded by the switch action as potential object locations in the image. It was also known that CUB contains a single object per image. Hence, based on the number of final predicted regions for a small subset of train images (based on approximately 30 samples), we tuned down the value of $switch\_loc\_th$ to 0.2. As a result, this forces FALcon to be more discriminative in terms of which foveated regions to consider as having potential object. Note that this hyperparameter remains the same for all evaluations on all other datasets, meaning that we did not infuse any more expert knowledge into the model.

Figure 8($a$) illustrates the distribution of the number of test images based on the number of predictions made by FALcon per image for CUB dataset. CUB dataset has a total of 5794 images in the test set. As it can be seen, without explicit supervision or forced restriction to make a single prediction per image, FALcon correctly predicts to make only 1 prediction for approximately 5500 samples (94.8%) and 2 predictions for 279 samples (4.8%). Only less than 1% of samples had more than 1 false positive predictions. Nevertheless, even when only 1 prediction is allowed, FALcon was able to achieve a significant localization performance on CUB dataset, which is notoriously known for its localization difficulty (Choe et al., 2023).

## B.2. ImageNet-1k dataset

The second dataset that FALcon was trained on is ImageNet-1k dataset, which is the classification and localization set of the ILSVRC2012 challenge. For this dataset, we used pre-trained PSOL localization model based on DenseNet151 (Huang et al., 2017) architecture, which is available at the official GitHub repository. For classification model, we used ResNet50 pre-trained on ImageNet-1k classification (image-level) labels.

FALcon localization model was implemented using VGG16 (Simonyan and Zisserman, 2015) DNN model. The model was trained using SGD optimizer with batch size of 512 samples, 0.9 momentum, 0.0001 weight decay for 100 epochs. Learning rate started from 0.01 and divided by 10 every 30 epochs. During training, FALcon hyperparameters were set to

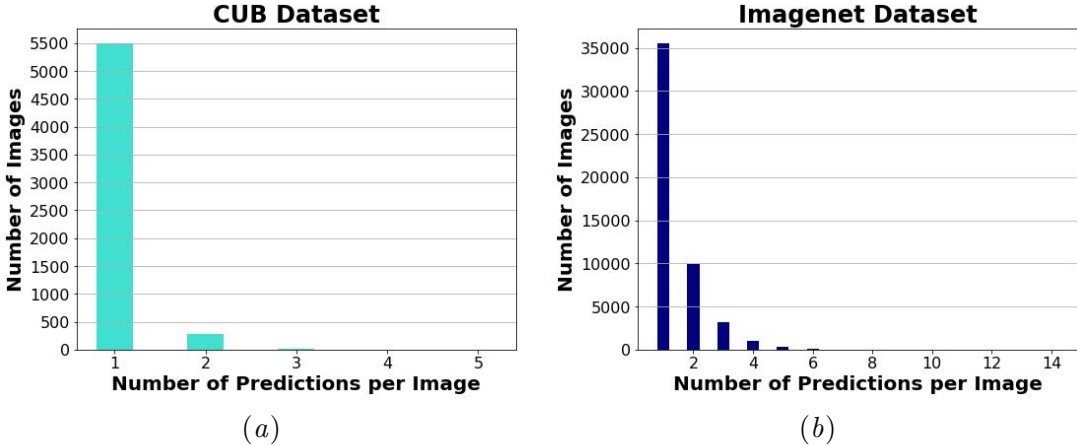

Figure 8: Distribution of the number of test images based on the number of predictions made by FALcon per image for (a) CUB dataset and (b) ImageNet-1k dataset.

match the ones used for CUB dataset: $num\_glimpses = 10$, $ratio\_wrong\_init\_glimpses = 0.5$, $glimpse\_size\_init = (20, 20)$, $glimpse\_size\_fixed = (96, 96)$, $glimpse\_size\_step = (20, 20)$, $glimpse\_change\_th = 0.5$, and $switch\_loc\_th = 0.5$. $num\_glimpses$ was reduced to 10 to fit the model during training into a single Nvidia A40 GPU card. During evaluation, we kept $switch\_loc\_th = 0.2$, similarly to CUB dataset, but used $glimpse\_size\_init = (20, 20)$. This also highlights that, due to the fact that FALcon only observes the input through resized foveated regions, it is capable of performing even when hyperparameters were transferred from the other dataset without extra fine-tuning.

Figure 8($b$) illustrates the distribution of the number of test images based on the number of predictions made by FALcon per image for ImageNet-1k dataset. ImageNet-1k dataset has a total of 50000 images in the test set. Similarly to CUB dataset, even without explicit supervision or forced limitation to make a single prediction per image, FALcon is able to narrow down the number of its predictions to 1 for 35518 samples (71%), to 2 for 9925 samples (19.85%), and to 3 for 3153 samples (6.3%). The remaining 3% of test samples have more than 3 predictions, with the outlier 8 samples getting more than 10 predictions per image. We believe the high number of false positives is attributed to the high diversity of images within ImageNet-1k dataset. Not only images purposefully contain images which have duplicate objects belonging to the same class, but also the natural background has high chance of containing objects which are not annotated but similar to annotated classes. Nevertheless, when we allow FALcon to be proactive and make up to 3 predictions per image, its localization performance improves significantly achieving competitive results with existing WSOL methods on ImageNet-1k dataset.

## Appendix C. Evaluation-only Experiments on Multi-object Images

The capabilities of FALcon on detecting multiple objects was evaluated on three datasets. The localization model trained on images from CUB dataset was evaluated on Pascal VOC07 and Pascal VOC12 datasets. Here the goal was to focus on (1) how many objects of class

"bird" from Pascal datasets will be detected by the models (both PSOL and FALcon) trained on CUB birds and (2) how well the FALcon model will be at not making unnecessary predictions. The model trained to localize objects of different classes from ImageNet-1k dataset was evaluated on the detection set of ILSVRC2013 challenge, which we refer to as ImageNet13-Det. It should be noted that only 150 out of 200 object classes in ImageNet13-Det are within 1000 classes of ImageNet-1k. Hence, we only considered FALcon predictions which had the predicted label as one of the 150 overlapping classes. Any other prediction was discarded, even if prediction had IoU greater than 0.5 with any ground truth box. When performing evaluation, we use the same set of hyperparameters that was used for the evaluation on their corresponding trained datasets. It means hyperparameters of CUB dataset were used for evaluations on Pascal VOC07 and VOC12 and of ImageNet-1k for evaluations on ImageNet13-Det.

## Appendix D. Additional WSOL Related Works

Excitation Backpropagation (Zhang et al., 2017) tackled the WSOL task by formulating a novel backpropagation scheme inspired by a top down human visual attention model. ADL Choe and Shim (2019) adversarially produces masks of high interests for localization. HaS (Singh and Lee, 2017) proposes a weakly supervised framework which unlike other contemporary WSOL techniques takes into account training the network on other relevant parts of the object when the most discriminative part is hidden. ACoL (Zhang et al., 2018b) showed how two parallel-classifiers are forced to leverage complementary object regions for classification to generate integral object localization together in an adversarial learning manner. C-WSL (Gao et al., 2018) improves upon WSOL performance by using a simple count-based region selection algorithm. During training, the goal is to select high-quality regions covering a single object instance. RCAM (Bae et al., 2020) introduced concepts such as Threshold Average Pooling (TAP) and Negative Weight Clamping (NWC) to alleviate the traditional CAM based approaches. Recently, BagCAMs (Zhu et al., 2022a) introduced a set of regional localizers from a pre-trained classifier to learn dedicated object-related spatial features to improve upon vanilla CAM approaches in an ensemble learning setup.

CutMix (Yun et al., 2019) introduces data augmentation strategy to learn relevant portions of various objects. DANet (Xue et al., 2019) proposes divergent activation functions to jointly learn complementary and discriminatory parts of the input. IVR (Kim et al., 2021) shows that different normalizations need to be used for different datasets and then introduces an novel Inferior Value Removal normalization strategy applicable for any CAM based WSOL technique. DA-WSOL (Zhu et al., 2022b) formulates the WSOL task as a domain adaptation problem and learns a source to target signal by localizing objects in the target domain via a source estimator. GC-Net (Lu et al., 2020) consists of generator that produces an object mask constrained by the location predicted by a detector describing a geometric shape.

BAS (Wu et al., 2022) argues that activation values are better for learning foreground prediction maps than cross-entropy and introduces an Activation Map Constraint module to suppress the background activation values to effectively localize the foreground object. CREAM (Xu et al., 2022) boosts activation values of integral regions of objects via a novel clustering approach that uses class specific foreground and background context embeddings

as cluster centroids. Alternatively, FAM (Meng et al., 2021) and ORNet (Xie et al., 2021) achieves the localization task by learning a prediction map of the foreground object. I2C (Zhang et al., 2020b) propose an inter-image communication strategy that keeps global consistency to learn class-specific features throughout the training set and a stochastic consistency to learn features of an object of the same category in the same mini-batch. This results to more robust localization maps. SPOL (Wei et al., 2021) uses suppressed shallow features in the background to learn sharp edges for enhanced localization. SPA (Pan et al., 2021) argues that structural information of an object is crucial for the localization task and proposes a two staged structure preserving activation to support the claim that includes a post processing feature extraction stage.

Since the advent of vision transformers Dosovitskiy et al. (2021) there have been quite a few transformer based approaches for WSOL by exploiting the attention scores for obtaining localization maps. LOST (Siméoni et al., 2021) leverages the learnt representations of a self-supervised pre-trained vision transformer to exploit the patch correlations within a single input image for object localization. Wang et al. (2022) formalizes the previous approach as a graph-cut algorithm to localize the foreground object. TokenCut utilizes the visual tokens as nodes in an unweighted graph with the edges representing a connectivity score based on similarity of tokens that ultimately leads to cutting out the foreground as the localized object. Both LOST and TokenCut use DINO (Caron et al., 2021) as the self-supervised pre-trained vision transformer. Bai et al. (2022) introduces a Spatial Calibration module (SCM) to generate attention maps that captures the sharper object boundaries and filters out the object-irrelevant background area. There have been other Transformer based WSOL works such as TS-CAM (Gao et al., 2021), ViTOL (Gupta et al., 2022), Re-Attention (Su et al., 2022) and GenPromp (Zhao et al., 2023).

