# OpenReview forum: "Exploring Foveation and Saccade for Improved Weakly-Supervised Localization"
_NeurIPS.cc/2023/Workshop/Gaze_Meets_ML — Gaze Meets ML 2023 Poster_

### Official Review · Reviewer_ngbE · 2023-10-12
**Biologically inspired approach to weakly supervised object localization**

**Rating:** 7
**Confidence:** 3

**Review:**

The authors propose a novel approach for weakly supervised object localization (WSOL) inspired by the human vision system.
Namely, the authors integrate saccades and foveation into a WSOL pipeline and demonstrate the effectiveness of their approach on common benchmark datasets against other WSOL approaches of the literature. Moreover, they show that the model exhibits some "human-like" classification and exploration properties using the Duck and Rabbit optical illusion as an example.

I find the approach to be interesting. However, in my opinion the authors should reduce this claim:
"Finally, using the interesting “duck-rabbit” optical illusion, we show that our method manifests human-like behavior."
The experiment is interesting, but it is limited to a single image and does not provide enough evidence for a generalized statement.

---

### Official Review · Reviewer_mr1q · 2023-10-21
**WSOL approach inspired by foveation and saccades**

**Rating:** 6
**Confidence:** 4

**Review:**

This paper presents a novel method for weakly supervised object localization (WSOL) that takes inspiration from the foveation and saccade features of the human visual system. The authors claim that the method offers performance benefits over existing methods, allows resilience in the form of in-built multi-object detection, and contributes to the neuro-inspired algorithm literature. Overall, I believe that the paper offers a new approach to WSOL that is beneficial for localizing multiple objects, with my main concerns being weaker quantitative results. There are several things that the authors do well:

* The paper is clearly written and generally well-organized.
* The authors are tackling an important problem, namely resiliency of vision models to complex stimuli with multiple objects.
* The authors are trying to incorporate features of human vision into their models. This is a young and growing effort, and overall important not only for performance and robustness but also for interpretability and human-model interaction.
* The authors do a very good job of contextualizing their work. Their literature review and discussions of prior work are excellent and very valuable for the reader.

My critiques and suggestions are the following:

* Results: From Table 1, DenseNet161 appears to outperform FALcon in across all selected metrics and datasets, which does not seem to be addressed. What do the authors see as the benefits of FALcon over DenseNet? Secondly, the authors claim that FALcon offers a significant improvement on the ImageNet dataset, pointing to a qualitative ability to detect multiple objects; however, there is barely an improvement in quantitative metrics. Perhaps the selected metrics do not accurately reflect the aspects of performance that the authors wish to capture. Fine-tuning the classification module, as the authors say, may also make for a better comparison.

* Evaluation: Are the selected evaluation metrics standard in the WSOL literature? Additionally, any interpretation of what the AP and mAP numbers mean would help contextualize the apparent benefits of FALcon. Finally, regarding "the object is considered to be correctly localized if any of the predictions correctly predicts the location and the object label" – although the authors later acknowledge the negative effect of this choice on precision, that effect is not reflected in the numerical metrics, which could be misleading.

* Methodology: I would consider moving the description of initial fixation generation (currently in 3.3) up to 3.1. Additionally, although explained later in the supplementary material, throughout the paper I was confused about how many initial fixation points there are, wondering what happens if no initial fixation is near the object target. I would also move the explanation and justification of PSOL and pseudo bounding boxes (currently in 3.3) up to 3.2, as it is hard to understand the training approach without knowing what the supervision is based on.

* Neuro-inspiration, optical illusion: First, in 4.4, please clarify what dataset the FALcon model was trained on before testing on this illusion. Second, a lot more more work could be done to support the idea that FALcon behaves like a human visual system. To me, the observed behavior of FALcon is unsurprising due to the particular construction of the illusion image (a real duck bill image attached to a real rabbit face image). The interesting thing about this illusion in humans is not that we base our judgement of duck or rabbit on the part of the image we foveate; it is that our perception of the entire image flips, regardless of where our initial or current fixation lands (known as perceptual bistability). What will FALcon do with an initial fixation that is in-between the duck and rabbit parts? Since this section is meant to support a major supposed benefit of FALcon, it could use a lot more work beyond application to a single instance of a particular optical illusion.

* Neuro-inspiration, F&S: I believe that it should be made clear what is and is not biologically inspired, as the methodology includes many key choices that are not. For example, the algorithm can only "saccade" between a pre-determined grid of fixations; the "fovea" is rectangular; there is no bound on how large the foveation window can get, with no corresponding loss in perceptual acuity; and the central approach of starting with a fixed-size initial foveation and expanding its size based on judged "interest". To my understanding, only the broad concepts of "seeing part of an image" and "changing focus area" are biological. Since "neuro-inspired" is a central claim of this paper, it should be explicitly stated whether or not a design choice is biologically inspired. Although, I come from a cognitive science background, so perhaps this is level of description is standard in "neuro-inspired" machine learning.

---

### Official Review · Reviewer_GStH · 2023-10-24
**Interesting idea, reasonable framework, and well-designed experimental setup. Results are underwhelming but consistent with other bio-inspired ML modification work.**

**Rating:** 6
**Confidence:** 5

**Review:**

#Review for "Exploring Foveation and Saccade for Improved Weakly-Supervised Localization"
##Fit
This paper is a good fit for the workshop and is relevant to the workshop goals.

## Quality
The quality of the study procedure, experimental design is above average [6/10]. The quality of the evaluation procedure is average [5/10]. The quality of the writing, visualization and paper presentation is medium high [8/10].

## Originality
The framework and application are both novel for the task.

## Significance and Impact
This work is a significant proof of concept towards more human-like DNNs that consider human visual processes and context for identifying salient features in images. The minor improvements observed in the evaluations is a deterrent for immediate impact. I think this work is important as a framework for future improvements.

## Clarity and Writing
The study procedure is clear, albeit a bit fragmented between the main body and appendix. There are a few formatting with whitespace and writing issues such as words and word trails bleeding onto the next line. These writing issues could be fixed and more space could be made to include important information that is currently in the appendix such as Appendix B.2 results,  to help format a better paper structure.

## Pros
- Bio-inspired ML augmentation has found success in various fields and this paper is an important proof-of-concept for human-like DNNs.
- Designed to consider multiple objects in the image by default without the need for ground truth boxes.
- Popular open datasets for object detection are used, making the results reproducible and techniques re-implementable for future work.
- The instances in which this approach shines, and why it works is discussed in proper detail in Section 5 and the appendix.


## Cons
- Some writing issues that can be easily handled before the presentation.
- The authors claim that their approach handles optical illusions in a manner consistent with human visual processes but do not show any empirical evidence that the performance is any better or worse than baselines that do not use foveal data.
- Underwhelming improvements when compared to DenseNet161, as well as the augmented baseline PSOL over ImageNet-1k dataset are the biggest cons. However, I understand that other bio-inspired works that augment existing DNNs with foveation and gaze data have seen similar 1-4% improvements.

---

### Meta-Review · Area_Chair_T7WT · 2023-10-26

**Recommendation:** Accept (Poster)
**Confidence:** 3

**Metareview:**

This paper proposes a new approach that combines foveation and saccades, inspired by human vision, to improve object localization, resulting in better bounding box predictions and the ability to detect multiple objects from single-object data. This method mimics human vision processes and exhibits human-like behavior.

The paper is well-written and presents a novel approach. However, the reviewers have raised concerns about the experimental aspects, particularly regarding optical illusions. The authors should consider narrowing the scope of their claims in the paper. While the experiment is intriguing, it has limitations due to its focus on a single image and the specific construction of the illusion used. Furthermore, it's essential to clearly delineate what aspects of the methodology are biologically inspired, as there appear to be significant design choices that deviate from this principle. Additionally, the improvement margin is minor, and the authors should discuss the reasons in the paper.

---

### Decision · Program_Chairs · 2023-10-26

Accept (Poster)